# *Acinetobacter thutiue* sp. nov. Isolated from Oil-Contaminated Soil in Motorbike Repair Workshops

**Nhan Le Thi Tuyet and Jaisoo Kim ***

Department of Life Science, College of Natural Sciences, Kyonggi University, Suwon 16227, Gyeonggi-Do, Republic of Korea; nhanle@kyonggi.ac.kr
* Correspondence: jkimtamu@kgu.ac.kr; Tel.: +82-31-249-9648

**Abstract:** Two *Acinetobacter* isolates were found in soil samples from motorbike repair workshop floors in Vietnam. The two *Acinetobacter* isolates were Gram-stain-negative, coccobacilli-shaped, aerobic, non-motile, non-hemolytic, oxidase-negative, and catalase-positive. They were designated as strains VNH17[T] and VNK23. Their growth was inhibited by NaCl concentrations of >3%, and they grew at an optimal temperature of 20–35 °C. Acidification of D-glucose and hydrolysis of gelatin were negative. They grew on β-alanine, ethanol, L-histidine, D-malate, and L-ornithine whereas L-arginine, citrate, L-glutamate, and L-phenylalanine were not utilized. Core genome-based phylogenetic analysis revealed that the two isolated strains formed a lineage within the genus *Acinetobacter* in the family *Moraxellaceae*, the closest relative was *Acinetobacter pavus* (91.70–91.95%), and were grouped within a hemolytic clade with other closely-related relatives. The DNA G+C content of VNH17[T] and VNK23 was 42.07% and 41.75%, respectively. The average nucleotide identity and in silico DNA-DNA hybridization relatedness values (74.41–91.61% and 20.6–45.40%, respectively) between the *Acinetobacter* isolates and phylogenetically related type strains were below the threshold values used for species delineation. Based on genomic, chemotaxonomic, phenotypic, and phylogenomic analyses, the isolated strains represent novel species in the genus *Acinetobacter*, for which the name *Acinetobacter thutiue* sp. nov. (type strain VNH17[T] = KACC 23003[T] = CCTCC AB 2023063[T]) is proposed.

**Keywords:** *Acinetobacter thutiue*; oil-contaminated soil; novel species; *Morazellaceae*





## 1. Introduction

The genus *Acinetobacter* [1] is a member of the family *Moraxellaceae*, with its type species being *Acinetobacter calcoaceticus*, which was isolated and described by Beijerinck in 1910 under the name *Micrococcus calcoaceticus* [2]. Currently, 80 species with correct, validly published names are assigned to the genus *Acinetobacter* (https://lpsn.dsmz.de/genus/acinetobacter; date of access: 26 October 2023) [3]. This genus comprises bacteria that are Gram-negative coccobacilli that are non-motile, non-spore-forming, aerobic, oxidase-negative, catalase-positive, and grow well on simple media. *Acinetobacter* species have been isolated from a variety of animal food products, including milk and meat [4–6]. Some *Acinetobacter* species have also been isolated from clinical sources [7,8] and floral nectar [9], whereas other members of this genus are widely distributed in soil and water [10–12]. Because of their versatile metabolic capabilities and the ability to degrade various compounds such as long-chain dicarboxylic acids and aromatics, some members of *Acinetobacter* can grow well under extreme environmental conditions, such as organic solvents and oil-contaminated soil [13–16]. This study aimed to characterize two *Acinetobacter* isolates, VNH17[T] and VNK23, found in oil-contaminated soils in Vietnam, and delineate their taxonomic position, which is considered to represent a novel species of the *Acinetobacter* genus.

## 2. Materials and Methods

### 2.1. Isolation and Ecology

During the process of isolating uncultured oil-degrading bacteria from oil-contaminated soils, a number of bacterial strains were isolated from soil samples of motorbike repair workshop floors in Vietnam (Figure S1). Among them, two strains, VNH17[T] and VNK23, were isolated from motorbike repair workshops at Phu Vang, Thua Thien Hue, Vietnam (17°04′16.3″ N 106°59′28.4″ E) and Dak Bla, Kon Tum, Vietnam (14°20′43.7″ N 108°04′48.9″ E), respectively. A modified culture method with a six-Transwell plate (Corning Inc., Corning, NY, USA) and mineral salt medium (MSM)-oil (2.42 g $KH_2PO_4$, 5.6 g $K_2HPO_4$, 2 g $(NH_4)_2SO_4$, 0.3 g $MgSO_4 \cdot 7H_2O$, 0.04 g $CaCl_2 \cdot 2H_2O$), 1 mL trace elements solution [4.5 g $MnSO_4 \cdot 7H_2O$, 0.1 g $CuSO_4 \cdot 5H_2O$, 0.1g $FeSO_4$, pH adjusted to 7.0 [17]], and 2 mL of crude oil per liter distilled water was used to culture the isolates. Soil samples (3 g) were kept on the bottom of the Transwell plates and 3 mL of MSM-oil was added, and 100 μL of the soil suspension was added to the insert. The Transwell plate was incubated in a shaker at 130 rpm at 28 °C for 4 weeks. After incubation, 100 μL of each diluted culture solution was spread onto an MSM-oil agar plate. Colonies were selected and streaked separately on MSM-oil agar plates until pure colonies were grown. The strains were then subcultured on R2A and stored at −70 °C in R2A broth supplemented with 20% (*v/v*) glycerol. *Acinetobacter junii* KACC 12228[T], *Acinetobacter parvus* KACC 12455[T], *Acinetobacter indicus* KCTC 42000[T], *Acinetobacter bereziniae* KCTC 42001[T] and *Acinetobacter venetianus* DSM 23050[T] were obtained from the Korean Agricultural Culture Collection (KACC), the Korean Collection for Type Cultures (KCTC), and Deutsche Sammlung von Mikroorganismen und Zellkulturen (DSMZ), and were used as the reference organisms in this study. Additionally, the following closely related species were used for comparison of physiological traits: *A. vivianii*, *A. courvalinii*, *A. proteolyticus*, *A. modestus*, *A. tjernbergiae*, *A. colistiniresistens*, *A. disperses*, *A. gyllenbergi*, *A. venetianus*, *A. haemolyticus*, and *A. halotolerans*.

### 2.2. Physiology and Chemotaxonomy

The colony morphologies of the VNH17[T] and VNK23 strains were observed after cultivation on R2A agar plates at 35 °C for two days. Cell morphology was examined using light microscopy (BX50; Olympus, Tokyo, Japan) and transmission electron microscopy (Bio-TEM H-7650; Hitachi, Tokyo, Japan). Gram staining was performed using the Hucker's method [18]. Next, endospore formation was investigated by staining the bacterial cell samples with malachite green as described by Schaeffer and Fulton [19]. Hydrogen sulfide and indole production and motility was assayed on sulfide indole motility (SIM) medium (CM0435; Oxoid, Hampshire, UK) after 48–72 h of incubation at 30 °C. Kovacs's reagent was then used to evaluate indole production [18]. Oxidase activity was assayed using 1% (*w/v*) tetramethyl-p-phenylenediamine. Catalase activity was determined based on bubble production after mixing a pellet of a fresh culture with a drop of 3% (*v/v*) hydrogen peroxide ($H_2O_2$). The test of growth on different agar media and temperature-dependent growth was performed by suspending fresh biomass in 0.9% (*w/v*) sodium chloride (NaCl). Next, the cell suspension was spotted on agar media and growth was monitored after 24, 48, and 72 h of incubation. Conventional media including R2A agar (MB Cell), tryptone soya agar (TSA; MB Cell), Luria Bertani (LB) agar (Oxoid), nutrient agar (NA; Oxoid), Mueller–Hinton agar (MB Cell), and MacConkey agar (MB Cell) were used. Temperature-dependent growth was determined at different temperatures (4, 10, 15, 20, 25, 28, 30, 32, 35, 37, 39, 41, 43, and 45 °C) on R2A agar. Growth physiology at different pH values was determined at 30 °C in R2A broth adjusted to pH 3–12 (increments of 0.5 pH units) using citrate/$NaH_2PO_4$ buffer (for pH 3.0–5.5), Sorensen's phosphate buffer (for pH 6.5–8), Tris buffer (for pH 8.5–9), carbonate buffer (for pH 9.5–10.0), and 5 M NaOH (for pH 10.5–11). Tolerance tests at different NaCl concentrations were performed in R2A supplemented with NaCl concentrations ranging between 0 and 7.0% (0.5% intervals) after five days of incubation at 30 °C. Cellular growth at various pH values and salinity levels was monitored by measuring optical density at 600 nm wavelength ($OD_{600}$) using a spectrophotometer (CARY 300, UV–Vis spectropho-

tometer; Varian, Palo Alto, USA). Substrate hydrolysis, including that of starch and casein, was assayed as described by Tindall et al. [20], Tween 80 was assessed according to the method of Smibert and Krieg [21], the Methyl Red and Voges-Proskauer (MR-VP) test was assayed with MR-VP broth (Vaughn et al., 1939), and the DNase test was performed on DNase agar (CM0321; Oxoid) and determined by observation after three days of incubation at 30 °C by flooding the plates with 1 M HCl. Nitrate reduction, indole production, gelatin hydrolysis, urease, aesculin, and other physiological and biochemical tests were performed using API 20NE, API 20E, and API ID 32 GN strips (bioMérieux, Marcy-l'Étoile, France) to evaluate basic chemical tests and carbon source utilization. The activities of various enzymes and acid formation from sugars were further determined using API ZYM test strips (bioMérieux). The commercial kits API 20NE, API 20E, API ID 32 GN, and API ZYM were used according to the manufacturer's instructions. Carbon source utilization was determined in test tubes filled with basal mineral medium supplemented with a 0.1% (*w/v*) carbon source according to established protocols [7]. The incubation temperature was 30 °C and results were assessed after 6 days of incubation. Antimicrobial susceptibility was analyzed by the disk diffusion method on Mueller–Hinton agar (Oxoid) at 30 °C. Antibacterial agents (μg/disk): piperacillin (100), ampicillin-sulbactam (10/10), gentamicin (10), tetracycline (30), and levofloxacin (5) were used, and the strains were classified into susceptible, intermediate, and resistant according to the recommendations of the CLSI (Clinical and Laboratory Standards Institute) [22]. To ascertain the presence of hemolytic activity, VNH17[T], VNK23, and reference strains were streaked on R2A agar supplemented with 5% (*v/v*) defibrinated sheep blood followed by incubation at 30 °C for five days.

To determine respiratory quinones and polar lipids, freeze-dried VNH17[T] and VNK23 cells that were grown on TSA and harvested after two days of growth were used. The respiratory quinones were extracted with methanol and petroleum ether from 70 mg freeze-dried cells, and quinone purification was performed according to the method of Minnikin et al. [23]. Purified quinones were identified by a reversed-phase high-pressure liquid chromatography (HPLC) system [solvent MeOH/isopropanol (7:5, *v/v*), flow rate 1.0 mL min$^{-1}$, wavelength 270 nm]. Polar lipids were extracted according Minnikin et al. [23] and separated by two-dimensional thin-layer chromatography (TLC) (Merck silica gel 60; $10 \times 10$ cm) and identified by spraying with appropriate detection reagents: 5% (*w/v*) ethanolic molybdatophosphoric acid (Sigma-Aldrich, St. Louis, USA) to detect of total lipids profiles, amino lipids were stained with 0.4% (*w/v*) solution of ninhydrin (Sigma-Aldrich) in butanol, phospholipids were detected with Zinzadze reagent (molybdenum blue spray reagent, 1.3%; Sigma Life Science), and glycolipids were detected with α-naphthol reagent (0.5%, *w/v*).

To profile cellular fatty acids, VNH17[T], VNK23, and the reference strains were grown on TSB agar and harvested after two days of growth. Cellular fatty acids were extracted, saponified, and methylated following the Sherlock Microbial Identification System version 6.3 (MIDI), analyzed by gas chromatography (GC), and identified using the TSBA6 database of the Microbial Identification System [24].

### 2.3. Phylogenetic Analysis Based on 16S rRNA Gene and Core Genome

The 16S rRNA gene sequences of strains VNH17[T] and VNK23 were obtained using polymerase chain reaction (PCR) with the primers 27F and 1492R, as described by Frank et al. [25]. The PCR products were sequenced and the 16S rRNA gene sequences were posted with the EzBioCloud database (www.ezbiocloud.net/eztaxon, accessed on 10 July 2023) [26] to identify the closest phylogenetic neighbors. The 16S rRNA gene sequences of the reference strains were retrieved from their depositions in the NCBI GenBank database (www.ncbi.nlm.nih.gov/, accessed on 10 July 2023). Multiple alignments of sequence data were performed using the SILVA aligner (www.arb-silva.de/aligner/ (accessed on 15 July 2023)). Three major phylogenetic trees (neighbor-joining [NJ], maximum-likelihood [ML], and maximum-parsimony [MP]) were constructed using mega version 7.0.26 [27].

The genomic DNA of VNH17$^T$ and VNK23 was extracted using DNeasy Blood and Tissue kits (Qiagen, Hilden, Germany). Whole-genome shotgun sequencing was performed by Macrogen (Seoul, Republic of Korea) using an Illumina HiSeq platform and assembled using SPAdes version 3.13.0 [28]. To analyze the evolutionary divergence based on whole-genome sequences, two phylogenetic trees of the two isolates and 15 main type strains or 79 all known type strains within the *Acinetobacter* genus from the NCBI (www.ncbi.nlm.nih.gov/ (accessed on 15 July 2023) were reconstructed in silico with the concatenated alignment of 92 core genes using the UBCG pipeline [29].

*2.4. Genome Features*

The DNA G+C content and average nucleotide identity (ANI) values between the whole genome sequences of the two isolates and the close reference strains were calculated by using the OrthoANIu (www.ezbiocloud.net/tools/ani (accessed on 3 August 2023) algorithm [30]. Digital DNA–DNA hybridization (dDDH) values between the VNH17$^T$ and VNK23 strains and between these and the type strains of the *Acinetobacter* genus were computed on the GGDC web server (http://ggdc.dsmz.de/ggdc.php (accessed on 4 August 2023) [31]. The genome sequences of VNH17$^T$ and VNK23 were annotated using the Rapid Annotation with the Subsystem Technology (RAST) server, version 2.0 [32]. To classify genes based on their function, the clusters of orthologous group (COG) analyses were performed by searching the Kyoto Encyclopedia of Genes and Genomes (KEGG) database [33].

## 3. Results and Discussion

### 3.1. Physiology and Chemotaxonomy

The two isolates shared a number of phenotypic characteristics, which are cell dimensions of 1.0–1.5 μm (length) and 0.8–1.0 μm (width), and coccobacilli-shaped (Figure S1). On R2A agar, VNH17$^T$ and VNK23 colonies were white, circular, convex and slightly opaque with entire margins, and had smooth surface, within 48 h at 30–35 °C. The VNH17$^T$ and VNK23 colony sizes were significantly different when cultured on R2A after 48 h incubation at 35 °C; the VNK23 colonies were 2–6 mm in diameter, whereas VNH17$^T$ colonies were notably smaller 0.1–0.5 mm (Figure S2). The two isolates grew well on R2A, TSA, NA, LB, Mueller–Hinton, and MacConkey agar. Furthermore, the strains exhibited the basic features of the genus *Acinetobacter*, that is, they were oxidase-negative, catalase-positive, Gram-stain-negative, and incapable of dissimilative denitrification [34].

Starch, Tween 80, casein, urease, aesculin, and gelatin hydrolysis and sheep blood hemolysis were negative. The inability to lyse sheep red blood cells was a trait shared with *A. parvus*, a known non-hemolytic species in the hemolytic clade [35], as was utilization of L-glutamate (Table 1). The two isolates grew at 0–3% (*w/v*) NaCl added in the medium; however, their growth was inhibited by NaCl concentrations of >3%. The pH range for growth was 6–10 (optimum: 6.5–7.5). The optimum temperature for growth was 20–35 °C, but the temperature rank for growth differed between the two isolates; VNH17$^T$ grew at 10–39 °C, but VNK23 did not grow at 10 °C for 5 days. Both isolates had no growth at temperatures of >41 °C within 14 days and did not anaerobically grow on R2A for five days at 30 °C. Both strains were susceptible to ampicillin-sulbactam, gentamicin (intermediate in VNH17$^T$), tetracycline, and levofloxacin. However, they showed different susceptibility profiles, with VNH17$^T$ being resistant to piperacillin, whereas VNK23 was susceptible to piperacillin (Table S1).

A detailed summary of the phenotypic features of the two isolated strains compared to closely related type strains is presented in Table 1 and Table S2. All strains shared many phenotypic similarities, indicating their relatedness and placement in the genus *Acinetobacter* while some differential properties of two isolated strains compared with the closest reference strains showed that two isolated strains were different from their closest neighbors (*A. parvus*, *A. vivianii*, *A. courvalinii*, *A. proteolyticus*, *A. modestus*, *A. tjernbergiae*, *A. colistiniresistens*, *A. disperses*, *A. gyllenbergii*, *A. junii*, *A. venetianus*, *A. haemolyticus*, and

*A. halotolerans*). None of *Acinetobacter* strains grew at ≥41 °C. Acidification of D-glucose, hydrolysis of gelatin, and hemolysis of sheep blood were negative. They grew on β-alanine, ethanol, L-histidine, D-malate, and L-ornithine whereas L-arginine, citrate, L-glutamate, and L-phenylalanine were not utilized. The only difference between the VNH17[T] and VNK23 strains from the closest species, *A. parvus*, was the utilization of L-histidine, which is shared with other hemolytic species in the clade (Table 1).

**Table 1.** Metabolic and physiological properties of *A. thutiue* sp. nov. and its phylogenetically closest related species. (1) *A. thutiue* sp. nov. (n = 2 where n is the number of strains); (2) *A. parvus* (n = 10); (3) *A. vivianii* (n = 9); (4) *A. courvalinii* (n = 9); (5) *A. proteolyticus* (n = 6); (6) *A. modestus* (n = 7); (7) *A. tjernbergiae* (n = 2); (8) *A. colistiniresistens* (n = 24); (9) *A. dispersus* (n = 9); (10) *A. gyllenbergii* (n = 9); (11) *A. junii* (14); (12) *A. venetianus* (5); (13) *A. haemolyticus* (16); (14) *A. halotolerans* (1). The results for *A. thutiue* sp. nov. were obtained from the present study, other species from Nemec et al. [36]. Assimilation tests were interpreted after 6 days of culture at 30 °C. +, All strains positive; -, all strains negative; D, (mostly) doubtful or irreproducible reactions; w, (mostly) weak positive reactions. Numbers are percentages of strains with positive reactions. For strain-dependent reactions, results for type strains are given in parentheses.

| Characteristic | 1 | 2 | 3 | 4 | 5 | 6 | 7 | 8 | 9 | 10 | 11 | 12 | 13 | 14 |
|---|---|---|---|---|---|---|---|---|---|---|---|---|---|---|
| Growth at 44 °C | - | - | - | - | - | - | - | - | - | - | - | - | - | - |
| Growth at 41 °C | - | - | - | 50W (D) | - | - | - | - | - | - | 93 (+) | - | 94 (+) | -! |
| Growth at 37 °C | + | 90 (+) | + | + | + | + (W) | - | 82 (+) | + | + | + | + | + | + |
| Growth at 35 °C | + | + | + | + | + | + | - | + | + | + | + | + | + | + |
| Growth at 32 °C | + | + | + | + | + | + | + | + | + | + | + | + | + | + |
| Acidification of D-glucose | - | - | 89 (+) | + | - | - | - | + | - | - | - | - | 75 (+) | + |
| Hemolysis of sheep blood | - | - | + | 89 (+) | + | + | + | + | + | + | 50 (-) | + | + | + |
| Liquefaction of gelatin | - | - | - | + | + | - | - | + | + | + | - | 80 (+) | 94 (+) | + |
| Utilization of Acetate | + | + | + | + | + | + | + | + | + | + | + | + | + | + |
| trans-Aconitate | - | - | 56 (-) | 44 (-) | + | - | - | - | 11 (-) | - | - | - | 63 (+) | + |
| Adipate | - | - | 89 (+) | + | 67 (-) | 14 (-) | - | - | 22 (-) | + | - | 20 (-) | - | - |
| β-Alanine | + (W) | - | + | 89 (+) | + | - | - | 17 (-) | + | + | - | - | - | - |
| 4-Aminobutyrate | - | - | + | + | D (-) | - | - | - | + | D | 86 (+) | + | + | + |
| L-Arabinose | - | - | - | - | - | - | - | - | - | - | - | - | - | - |
| L-Arginine | - | - | + | + | + | + | 50 (+) | + | + | + | 93 (+) | + | 94 (+) | + |
| L-Aspartate | - | - | 11 (D) | 11 (-) | - | - | - | - | - | - | 21 (+) | - | 31 (-) | + |
| Azelate | - | - | 89 (+) | + | 67 (-) | - | - | - | 22 (-) | + | - | 20 (-) | - | - |
| Benzoate | 50 (+) | - | + | + | + | + | - | 17 (-) | + | + | 79 (+) | + | - | - |
| 2,3-Butanediol | - | - | - | - | - | 14 (-) | - | - | - | - | - | - | - | - |
| Citraconate | - | - | - | - | - | - | - | - | - | - | - | - | - | - |
| Citrate (Simmons) | - | - | + | + | + | - | - | + | + | + | 79 (+) | + | 75 (+) | + |
| Ethanol | + | + | - | - | - | + | + | 17 (-) | 11 (-) | 22 (-) | 93 (+) | + | 94 (+) | + |
| Gentisate | - | - | 89 (+) | - | + | - | - | 88 (+) | 33 (-) | 11 (+) | - | - | 81 (+) | - |
| D-Gluconate | - | - | - | - | - | - | - | - | - | - | - | - | - | - |
| D-Glucose | - | - | - | - | - | - | - | - | - | - | - | - | - | - |
| L-Glutamate | - | - | + | + | + | + | + | + | + | + | + | + | + | + |
| Glutarate | - | - | 89 (+) | + | 33 (-) | - | - | - | + | D (-) | - | - | - | - |

**Table 1.** *Cont.*

| Characteristic | 1 | 2 | 3 | 4 | 5 | 6 | 7 | 8 | 9 | 10 | 11 | 12 | 13 | 14 |
|---|---|---|---|---|---|---|---|---|---|---|---|---|---|---|
| Histamine | - | - | - | 11 (-) | - | - | - | - | - | - | - | - | - | - |
| L-Histidine | + | - | + | + | + | + | + | + | + | + | 93 (+) | + | + | + |
| 4-Hydroxybenzoate | D | - | + | + | + | - | - | 83 (+) | + | 89 (+) | - | - | 81 (+) | + |
| DL-Lactate | + (W) | - | + | + | + | - | - | 96( +) | + | + | 93 (+) | - | - | - |
| L-Leucine | - | - | + | + | 83 (-) | - | - | 13 (-) | + | + | 14 (-) | + | 88 (+) | + |
| Levulinate | - | - | - | 11 (-) | - | - | - | - | - | - | - | - | - | - |
| D-Malate | + | - | + | 89 (+) | + | D | - | 92 (+) | + | + | 79 (D) | + | 88 (+) | + |
| Malonate | - | - | + | + | 67 (+) | - | - | 8 (-) | 22 (-) | 78 (+) | - | + | - | - |
| L-Ornithine | + (W) | 20 (-) | + | 89 (+) | + | - | - | - | 89 (D) | 56 (+) | - | - | - | - |
| Phenylacetate | D | - | + | + | 83 (-) | - | - | 96 (+) | 89 (+) | + | - | - | - | - |
| L-Phenylalanine | - | - | + | + | + | - | - | 96 (+) | 89 (+) | 89 (+) | - | - | - | - |
| Putrescine | - | - | 78 (+) | + | - | - | - | - | + | - | - | - | - | - |
| D-Ribose | - | - | 22 (+) | - | - | - | - | - | - | - | - | - | - | - |
| L-Tartrate | - | - | - | 22 (-) | - | - | - | - | - | - | - | - | - | - |
| Tricarballylate | - | - | 56 (-) | 44 (-) | + | - | - | - | 11 (-) | - | - | - | - | - |
| Trigonelline | - | - | + | 67 (+) | - | + | 50 (+) | 33 (-) | - | - | - | - | - | - |
| Tryptamine | - | - | D | 22 (D) | 50 (+) | - | - | - | 11 (+) | - | - | - | - | - |

Respiratory quinones of the VNH17$^T$ and VNK23 strains were ubiquinone 9 (Q-9: 75.79% and 74.41%, respectively), ubiquinone 8 (Q-8: 22.41% and 23.67%, respectively), and small amounts of Q10 (Q-10: 1.8% and 1.92%, respectively) (Figure S4). Polar lipids included diphosphatidylglycerol (DPG), phosphatidylglycerol (PG), phosphatidylethanolamine (PE), and an unidentified phospholipid (PL). No glycolipids were detected (Figure S5). Similar to those of the reference strains of members of the genus *Acinetobacter*, the prime fatty acids of the VNH17$^T$ and VNK23 strains were $C_{18:1}\omega9c$ (49.42% and 36.56%, respectively), $C_{16:0}$ (18.96% and 22.53%, respectively), $C_{12:0}3OH$ (3.79% and 4.18%, respectively), $C_{12:0}$ (2.92% and 3.34%, respectively), and summed feature 3 ($C_{16:1}\omega7c$ and/or $C_{16:1}\omega6c$; 16.38% and 25.89, respectively). However, VNH17$^T$ had the presence of small amounts of anteiso–$C_{17:0}$ (0.17%), $C_{14:1}$ $\omega5c$ (0.1%), and $C_{18:3}$ $\omega6c$ (0.11%) which were not detected in VNK23 (Table 2). The fatty acid composition of VNH17$^T$ and VNK23 was consistent with previous results for recognized species of the genus *Acinetobacter*.

**Table 2.** Detailed cellular fatty acid profiles (% of totals) of the VNH17$^T$, VNK23 strains and closely related reference strains. Strains: (1) VNH17$^T$; (2) VNK23; (3) *Acinetobacter parvus* KACC12455$^T$; (4) *Acinetobacter juni* KACC12228$^T$; (5) *Acinetobacter indicus* KCTC42000$^T$; (6) *Acinetobacter bereziniae* KCTC42001$^T$; (7) *Acinetobacter venetianus* DSM23050$^T$. All data are from the present study. Fatty acids that represent <0.1% of the total in all strains are not shown; -, <0.1% or not detected.

| Fatty Acid | 1 | 2 | 3 | 4 | 5 | 6 | 7 |
|---|---|---|---|---|---|---|---|
| Saturated | | | | | | | |
| $C_{10:0}$ | 0.56 | 0.57 | 0.33 | 1.35 | 0.13 | - | 1.35 |
| $C_{11:0}$ | - | - | - | 0.11 | - | - | - |
| $C_{12:0}$ | 2.92 | 3.34 | 6.21 | 2.80 | 6.55 | 9.31 | 5.18 |
| $C_{14:0}$ | 0.98 | 1.19 | 0.52 | | 1.27 | 0.81 | 0.78 |
| $C_{16:0}$ | 18.96 | 22.53 | 9.02 | 15.59 | 14.05 | 20.46 | 24.43 |

**Table 2.** *Cont.*

| Fatty Acid | 1 | 2 | 3 | 4 | 5 | 6 | 7 |
|---|---|---|---|---|---|---|---|
| N alcohol $C_{16:0}$ | - | - | - | 0.14 | - | 0.17 | - |
| $C_{17:0}$ | 0.19 | 0.19 | 0.37 | 0.92 | 0.22 | - | 0.29 |
| Iso $C_{17:0}$ | - | | 0.23 | 1.04 | - | - | - |
| $C_{18:0}$ | 2.12 | 0.33 | - | 1.68 | 1.85 | 1.19 | 0.97 |
| 10-methyl $C_{17:0}$ | - | - | - | - | - | - | - |
| anteiso–$C_{17:0}$ | 0.17 | - | - | - | - | - | - |
| Unsaturated | | | | | | | |
| $C_{14:1}$ ω5c | 0.10 | - | - | - | - | - | - |
| $C_{16:1}$ ω5c | - | - | - | 0.10 | - | - | - |
| $C_{16:1}$ ω7c alcohol | - | - | - | - | - | - | - |
| $C_{16:1}$ ω9c | - | - | - | 0.9 | - | - | - |
| $C_{17:1}$ ω8c | 0.60 | 0.77 | 2.49 | 2.86 | 0.58 | 0.22 | 1.22 |
| $C_{18:1}$ ω9c | 49.42 | 36.56 | 39.37 | 38.41 | 29.76 | 30.14 | 29.68 |
| $C_{18:1}$ ω5c | - | - | - | - | - | - | 0.11 |
| $C_{18:3}$ ω6c | 0.11 | - | - | 0.22 | - | 0.21 | - |
| Iso I $C_{19:1}$ | 0.14 | 0.31 | - | - | - | 0.18 | 0.17 |
| Hydroxy | | | | | | | |
| $C_{8:0}$ 3OH | - | - | 0.13 | - | - | - | - |
| C12:0 2OH | 2.15 | 2.33 | 3.63 | 2.89 | 0.57 | 0.63 | 2.25 |
| C12:0 3OH | 3.79 | 4.18 | 6.73 | 4.75 | 3.04 | 5.59 | 5.19 |
| Summed features * | | | | | | | |
| Summed Feature 1 | - | - | - | 0.13 | - | - | - |
| Summed Feature 2 | 0.14 | 0.24 | 0.28 | 0.3 | 2.61 | 0.10 | 0.22 |
| Summed Feature 3 | 16.39 | 25.89 | 29.75 | 23.70 | 36.84 | 28.41 | 26.90 |
| Summed Feature 8 | 1.18 | 1.48 | 0.94 | 0.89 | 2.49 | 2.40 | 1.17 |

\* Summed features represent two or three fatty acids that cannot be separated by the Microbial Indentification System. Summed feature 1 consisted of $C_{15:1}$ iso H/$C_{13:0}$ 3OH, summed feature 2 consisted of $C_{16:1}$ iso I/$C_{14:0}$ 3OH; summed feature 3 consist of $C_{16:1}$ ω7c/$C_{16:1}$ ω6c; summed feature 8 consisted of $C_{18:1}$ ω7c/$C_{18:1}$ ω6c.

*3.2. Phylogenetic Analysis Based on 16S rRNA Gene and Core Genome*

16S rRNA gene analysis revealed that VNH17[T] (1468 nucleotides) and VNK23 (1472 nucleotides) were separated into a cluster in a phylogenetic tree in the genus *Acinetobacter* of the family *Moraxellaceae* (Figure S6). The close relationship between the two organisms was reflected by the 100% bootstrap support and the topology of the NJ tree was supported by those of the ME (Figure S7) and ML trees (Figure S8). Strain VNH17[T] was most closely related to *Acinetobacter modestus* (96.98%) and VNK23 was most closely related to *Acinetobacter junii* (97.26%). To verify this result, we further analyzed their genome features.

However, core genome-based phylogeny revealed that the two strains (VNH17[T] and VNK23) were most closely related to *Acinetobacter pavus* (91.70% and 91.95%, respectively) and distantly grouped together with *A. modestus*, *A. tjernbergiae*, *A. courvalinii*, *A. vivianii*, *A. dispersus*, *A. colistiniresistens*, *A. proteolyticus*, and *A. gyllenbergii* as a clade (Figure 1 and Table S3). Current trends have shown that 16S rRNA gene-based phylogeny and similarity are unreliable for selecting the phylogenetically closest relatives of new lineages, so genome-wide comparisons should be useful for key criteria [37].

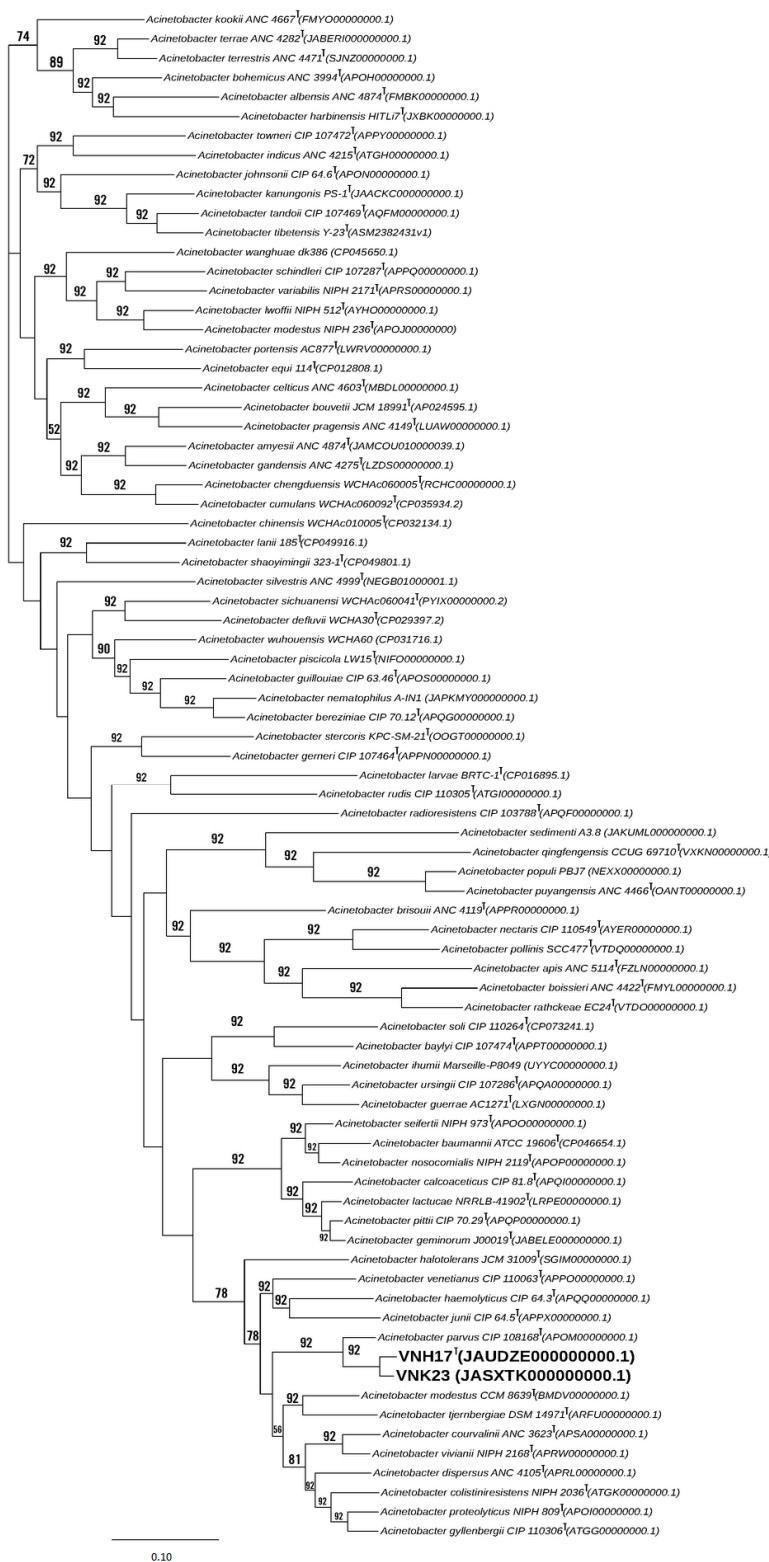

**Figure 1.** The core genome-based phylogenetic tree was reconstructed using the UBCG (concatenated alignment of 92 core genes) showing the phylogenetic placement of the two strains of VNH17[T] and VNK23 (two strains of novel species *Acinetobacter thutiue*) and including all known *Acinetobacter* type strains with validly published names. GenBank accession numbers are given in parentheses. Scale bar, 0.1 nucleotide substitutions per site.

*3.3. Genome Features*

The complete genome of strain VNH17[T] was 3873739 bp in size, the N50 of the filtered reads was 331771 kb, the genome coverage was 56.68×, and the DNA G+C content was 42.1 mol%. For VNK23, 3947147 bp was the size of the complete genome, the N50 of the filtered reads was 174689 kb, the genome coverage was 55.73×, and the DNA G+C content was 41.8 mol%. RAST analysis revealed the presence of 3721 coding sequences, 72 RNAs, and 291 subsystems in the genome in strain VNH17[T] For VNK23 there were 3926 coding sequences, 77 RNAs, and 289 subsystems. A total of 3586 protein-coding genes were identified in VNH17[T], of which 3461 (96.5%) were assigned to COG categories; these numbers for VNK23 were 3722 and 3525 (94.7%), respectively. The COG category assignment of genes to general function prediction of VNH17[T] and VNK23 was 13.5% (468 genes) and 14.0% (495 genes), and other known functions included amino acid transport and metabolism 5.0% (175 genes) and 4.7% (166 genes), inorganic ion transport and metabolism 6.1% (211 genes) and 5.3% (186 genes), and energy production and conversion 4.45% (154 genes) and 4.3% (152 genes), 28.3% and 28.5% of genes functioned unknown into the COGs, respectively (Figure S9). Pairwise ANI and dDDH values between the VNH17[T] and VNK23 strains were 96.05 and 65.70%, respectively, the pairwise ANI values between them and other type strains of the genus *Acinetobacter* tested ranging from 74.41 to 91.61%, and dDDH values ranged from 20.6 to 45.40% (Table S3). In light of the recommended threshold values of ANI 95–96% [38] and dDDH 70% [32] for species circumscription, these values indicated that the ANI value between VNH17[T] and VNK23 was higher than the generally accepted the species boundary, but the dDDH value was within the accepted species boundary. These values indicate that the two isolated strains are clearly separated from all known taxa at the species level. However, the two isolated strains shared many phenotypic features and some *Acinetobacter* species showed an interspecies range of dDDH between 64.7 and 99.3% [39]. Therefore, the two isolated strains need to be subjected to further genomic and phenotypic analyses.

**4. Conclusions**

Phylogenetic analysis identified the VNH17[T] and VNK23 isolates as members of the *Acinetobacter* genus. Both isolates could be clearly differentiated from closely related *Acinetobacter* species based on the isolation source, growth characteristics, enzymatic activities, assimilation patterns, and genetic analysis. Furthermore, the shape and size of colonies, and rank temperature for growth were distinct. Thus, the VNH17[T] and VNK23 strains represent a novel species of the genus *Acinetobacter*, for which the names *Acinetobacter thutiue* sp. nov. has been proposed. Future research should aim to clarify whether *Acinetobacter thutiue* sp. nov. has any relevant role in oil degradation to elucidate its importance for hydrocarbon degradation capacity, its growth properties in oil-polluted environments, and its significance in oil-contaminated soil ecosystems.

*Description of Acinetobacter thutiue sp. nov.*

*Acinetobacter thutiue* (*thu.ti.ue.* N.L. neut. adj. thutihue from Thua Thien Hue, the region of Vietnam from where the type strain was isolated).

The cells are Gram-negative, aerobic, oxidase-negative, catalase-positive, non-motile, and coccobacilli. Colonies on R2A after 48 h incubation at 35 °C are approximately 0.1–0.5 mm in diameter, circular, convex, smooth, and milky-white in color. The cells grow at a temperature of 10–39 °C (optimum, 20–35 °C) and pH 6.0–10.0 (optimum pH, 6.5–7.5). Starch, Tween 80, casein, urease, aesculin, and gelatin hydrolysis and sheep blood hemolysis are negative. Acid is not produced by D-glucose. The following substrates are utilized as the sole carbon sources: acetate, ethanol, L-Histidine, glutarate, D-malate, and benzoate. No growth occurs on trans-aconitate, adipate, 4-aminobutyrate, l-arabinose, l-arginine, l-aspartate, azelate, 2,3-butanediol, citraconate, citrate, gentisate, d-gluconate, d-glucose, L- glutamate, glutarate, histamin, l-leucine, levulinate, malonate, l-phenylalanine, putrescine, d-ribose, l-tartrate, tricarballylate, trigonelline, and tryptamine.

The type strain is VNH17$^T$ (=KACC 23003$^T$ = CCTCC AB 2023063$^T$), isolated from the oil-contaminated soil of motorbike repair workshop floors (Thua Thien Hue province, Vietnam). The genome is 3,873,739 bp in size, with a G+C content of 42.1 mol%. GenBank accession numbers for sequences of the strain *Acinetobacter thutiue* VNH17$^T$ are JAUDZE000000000 (genome) and OP727583 (16S rRNA gene nucleotide sequence).

**Supplementary Materials:** The following supporting information can be downloaded at: https://www.mdpi.com/article/10.3390/d15121179/s1, Figure S1: Sampling sites at the automobile workshops; Figure S2: Colonies of VNH17$^T$ and VNK23 were grown on R2A at 35 °C for 48 h; Figure S3: Transmission electron microscopy of the growth of the VNH17$^T$ and VNK23 strains on R2A medium plates for 2 days at 30 °C; Figure S4: Quinone analysis results of the VNH17$^T$ and VNK23 strains; Figure S5: Polar lipid profile of the VNH17$^T$ and VNK23 strains; Figure S6: The phylogenetic tree was reconstructed with the neighbor-joining method based on 16S rRNA gene sequences of the two isolated and type species of the genus Acinetobacter; Figure S7: Phylogenetic tree reconstructed with the Minimum Evolution method based on 16S rRNA gene sequences of two isolated and type species of the genus *Acinetobacter*; Figure S8: Phylogenetic tree reconstructed with the Maximum Likelihood method based on 16S rRNA gene sequences of the two isolated and type species of the genus *Acinetobacter*; Figure S9: COG functional classification of proteins in the VNH17$^T$ and VNK23 strains' genome; Table S1: Zone diameter breakpoint (mm) of antimicrobial agents for isolated bacteria; Table S2: Results from API ZYM, API 20NE test; Table S3: ANIb and dDDH values (%) between the genome sequences of the 16 strains in the genus *Acinetobacter*.

**Author Contributions:** N.L.T.T. conceived, designed, and conducted all of the experiments. N.L.T.T. and J.K. interpreted all of the data and read, discussed, edited, and approved the final draft of the manuscript. J.K. coordinated and supervised the study. All authors have read and agreed to the published version of the manuscript.

**Funding:** This work was supported by Kyonggi University's Graduate Research Assistantship 2020.

**Institutional Review Board Statement:** Not applicable.

**Data Availability Statement:** The data presented in this study are available in this published article and supplementary materials.

**Conflicts of Interest:** The authors declare no conflict of interest.

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
