# Peer review of "Acinetobacter thutiue sp. nov. Isolated from Oil-Contaminated Soil in Motorbike Repair Workshops"

_diversity, doi:10.3390/d15121179_

Round 1
Reviewer 1 Report (Previous Reviewer 2)
Comments and Suggestions for Authors
The manuscript is improved but I still have few comments
L32 - (https://lpsn.dsmz.de/genus/cellulomonas???
L174 - was annotated - were annotated
Figure 2 - VNH17 and VNK23 sequences lack GenBank accession numbers
Comments on the Quality of English Language
just few corrections
Author Response
Reviewer 1:
Comments and Suggestions for Authors
The manuscript is improved but I still have few comments
L32 - (https://lpsn.dsmz.de/genus/cellulomonas???
Answer: revised correctly
L174 - was annotated - were annotated
Answer: revised as suggested
Figure 2 - VNH17 and VNK23 sequences lack GenBank accession numbers
Answer: Figure 2 is omitted based on another reviewer’s comment

Reviewer 2 Report (Previous Reviewer 1)
Comments and Suggestions for Authors
The re-submitted paper has improved, but still suffers from a number of flaws and needs to be revised carefully.
Major comments
1. The taxonomy at and below the genus level is currently governed by two kinds of comparative analysis of whole genome sequences – calculations of pairwise similarities between genomes (OGRI) and core genome-based phylogeny. Any new nomenclatural proposal should define the taxonomic context of a novel species using these analyses. As the present paper is inconsistent in this regard, I recommend – in order to improve it - (i) to base the phylogeny of the novel strains primarily on core genome phylogram, (ii) omit completely the rpoB sections (a single gene marker guarantees nothing!) and (iii) reduce the 16S rRNA gene section given the well-known limitations of this marker below the genus level (both figures 1 and 2 are incomplete a contain errors!). Core genome-based phylogeny/similarity should be used as a primary criterion to select the phylogenetically closest relatives of the novel strains for detailed analyses.
2. Further, as being apparent from the presented data, the features of the novel strains should be compared mainly to biochemically less active members of the hemolytic clade (see PMID: 26822020, PMID: 28671519 and/or https://onlinelibrary.wiley.com/doi/10.1002/9781118960608.gbm01203.pub2), especially to A. parvus, with which they share the inability to lyse sheep blood (a very important phylogenetic signal!) and in one case, also the formation of small colonies (a unique feature of A. parvus within the whole genus!).
3. L196-7. If this was correct, it would be the first know Acinetobacter growing anaerobically. Furthermore, it is unlikely that strains of one species differ in such a pivotal and metabolically complex feature. The authors are requested to provide convincing data (incl. functional genomics-based) and arguments to support this extraordinary finding.
4. L8-26. Abstract should briefly summarize most relevant characteristics that distinguish the novel strains from the other member of the genus and avoid the features common for the whole genus Acinetobacter, e.g. by specifying which phenotypic and/or chemotaxonomic markers (the phenotypic ones have such potential) support the status of the strains as a novel species. The genomically closest species (A. parvus!) of the novel strains and the fact that they are members of the hemolytic clade should be mentioned. As there is clear evidence of the membership of the strains in Acinetobacter, the info in L12-4 is redundant.
Detailed comments
L28-30. Incorrect. Micrococcus calcoaceticus is neither included nor even mentioned in the 1954 paper of Brisou and Prévot. Thus they could not rename it!
L30-2. Confusion of names with “distinct” species. There are many synonyms on the list.
L32. An incorrect website address.
L41-4. It needs to be rephrased.
L47. “To verify the role of uncultured oil-degrading bacteria in...” is not clear, as no such activity was studied.
L58-61. Why as many as 4 weeks? It is an extremely long period for bacteria to grow. How long did it take bacteria to yield visible colonies on the MSM-oil agar medium?
L62-5. Why these species were included, especially A. tandoii, which is not related to the novel strains?
L91-2. It is generally recommended to use liquid media and a water bath, which allow for more precise control of temperature.
L191. The R2A medium contains some amount of NaCl, therefore the value 0 (w/v) is unrealistic. Rephrase.
L192-3. How were the optimum growth values defined? Using growth rate or colony size?
L194-5. It is difficult to conclude that bacterium does not grow at low temperature. The culture time must be specified.
L205. Specify the closest neighbors.
L212. dispersus, gyllenbergii
L233. tandoii
Comments on the Quality of English LanguageAcceptable, but with numerous typos.
Author Response
Reviewer 2:
Comments and Suggestions for Authors
The re-submitted paper has improved, but still suffers from a number of flaws and needs to be revised carefully.
Major comments
- The taxonomy at and below the genus level is currently governed by two kinds of comparative analysis of whole genome sequences – calculations of pairwise similarities between genomes (OGRI) and core genome-based phylogeny. Any new nomenclatural proposal should define the taxonomic context of a novel species using these analyses. As the present paper is inconsistent in this regard, I recommend – in order to improve it - (i) to base the phylogeny of the novel strains primarily on core genome phylogram, (ii) omit completely the rpoB sections (a single gene marker guarantees nothing!) and (iii) reduce the 16S rRNA gene section given the well-known limitations of this marker below the genus level (both figures 1 and 2 are incomplete a contain errors!). Core genome-based phylogeny/similarity should be used as a primary criterion to select the phylogenetically closest relatives of the novel strains for detailed analyses.
Answer: followed as your recommendation
- Further, as being apparent from the presented data, the features of the novel strains should be compared mainly to biochemically less active members of the hemolytic clade (see PMID: 26822020, PMID: 28671519 and/or https://onlinelibrary.wiley.com/doi/10.1002/9781118960608.gbm01203.pub2), especially to A. parvus, with which they share the inability to lyse sheep blood (a very important phylogenetic signal!) and in one case, also the formation of small colonies (a unique feature of A. parvus within the whole genus!).
Answer: included additional two related species showing negative hemolysis (see table 1)
- L196-7. If this was correct, it would be the first know Acinetobacter growing anaerobically. Furthermore, it is unlikely that strains of one species differ in such a pivotal and metabolically complex feature. The authors are requested to provide convincing data (incl. functional genomics-based) and arguments to support this extraordinary finding.
Answer: omitted description about anaerobic growth due to no genomic evidence
- L8-26. Abstract should briefly summarize most relevant characteristics that distinguish the novel strains from the other member of the genus and avoid the features common for the whole genus Acinetobacter, e.g. by specifying which phenotypic and/or chemotaxonomic markers (the phenotypic ones have such potential) support the status of the strains as a novel species. The genomically closest species (A. parvus!) of the novel strains and the fact that they are members of the hemolytic clade should be mentioned. As there is clear evidence of the membership of the strains in Acinetobacter, the info in L12-4 is redundant.
Answer: revised as suggested
Detailed comments
L28-30. Incorrect. Micrococcus calcoaceticus is neither included nor even mentioned in the 1954 paper of Brisou and Prévot. Thus they could not rename it!
Answer: revised correctly
L30-2. Confusion of names with “distinct” species. There are many synonyms on the list.
Answer: revised correctly
L32. An incorrect website address.
Answer: revised correctly
L41-4. It needs to be rephrased.
Answer: rephrased as suggested
L47. “To verify the role of uncultured oil-degrading bacteria in...” is not clear, as no such activity was studied.
Answer: rephrased to be clear
L58-61. Why as many as 4 weeks? It is an extremely long period for bacteria to grow. How long did it take bacteria to yield visible colonies on the MSM-oil agar medium?
Answer: uncultured bacteria are usually slow growers so taking longer time than cultured bacteria. Some uncultured bacteria become faster by repetitive sub-cultivation; three days to yield visible colonies on the MSM-oil agar medium
L62-5. Why these species were included, especially A. tandoii, which is not related to the novel strains?
Answer: removed A. tandoii and added more related species
L91-2. It is generally recommended to use liquid media and a water bath, which allow for more precise control of temperature.
Answer: it was tested through liquid medium but in a shaking incubator for better aeration
L191. The R2A medium contains some amount of NaCl, therefore the value 0 (w/v) is unrealistic. Rephrase.
Answer: rephrase as suggested
L192-3. How were the optimum growth values defined? Using growth rate or colony size?
Answer: decided the optimum growth values based on growth rate
L194-5. It is difficult to conclude that bacterium does not grow at low temperature. The culture time must be specified.
Answer: specified as suggested
L205. Specify the closest neighbors.
Answer: specified as suggested
L212. dispersus, gyllenbergii
Answer: revised as suggested
L233. tandoii
Answer: no need due to removal from entire comparisons as a reference

Round 2
Reviewer 2 Report (Previous Reviewer 1)
Comments and Suggestions for Authors
The manuscript has improved, but it is still in part inconsistent. Therefore, I offer the authors my last recommendations, incl. clarification of the meaning of some statements:
1. Although the term haemolytic clade is included in the abstract, it is neither mentioned nor defined in the main text. I could be done via quoting a relevant paper (PMID: 37889259, 36288213, 28671519 or 26822020). Moreover, it would be beneficial for a reader to indicate the 13 species included in the clade in Fig. 2, which belong to the bottom cluster of the tree (A. halotolerans trough A. gyllenbergii). This should be reflected also in Table 1 by adding missing species and omitting A. gandensis.
2. L21-31. Consider rephrasing: "The genus Acinetobacter (citation) is a member of the family Moraxellaceae, with its type species, Acinetobacter calcoaceticus, which was isolated and described by W. Beijerinck in 1910 under the name Micrococcus calcoaceticus [1]."
3. L31-33. Consider rephrasing: "Currently, 80 species with correct, validly published names are assigned to the genus Acinetobacter (https://lpsn.dsmz.de/genus/acinetobacter; date of access: 26 October 2023) [3]." Note that now, there are 81 such species, with the newest A. higginsii (PMID: 37889259) not included in the quoted list yet; but that can be ignored, as this new species is distinct from A. thutiue.
4. Fig. 1. I can hardly see a reason to include Fig. 1 (16S rRNA) in the main text given its limited value, inferiority to Fig. 2 and partial discordance with Fig. 2. It can be included in the supplementary data file if the authors consider it relevant. Alternatively, the authors can prepare a new tree with all members of the haemolytic clade and discuss differences; but I do not see its usefulness given the well-known unsatisfactory resolution of the 16S rRNA gene below the genus level.
5. L71, 199, 208, Table 1. A. gandensis is neither a member of the haemolytic clade nor a “closely related” species (see Fig. 2) and should be omitted and replaced by missing species of the haemolytic clade (see point 1 above).
Comments on the Quality of English LanguageTask of the editorial office.
Author Response
Comments and Suggestions for Authors
The manuscript has improved, but it is still in part inconsistent. Therefore, I offer the authors my last recommendations, incl. clarification of the meaning of some statements:
- Although the term haemolytic clade is included in the abstract, it is neither mentioned nor defined in the main text. I could be done via quoting a relevant paper (PMID: 37889259, 36288213, 28671519 or 26822020). Moreover, it would be beneficial for a reader to indicate the 13 species included in the clade in Fig. 2, which belong to the bottom cluster of the tree (A. halotolerans trough A. gyllenbergii). This should be reflected also in Table 1 by adding missing species and omitting A. gandensis.
Answer: included the term haemolytic clade in the main text and all other species in the clade in Fig. 2 (now Fig. 1), as well as omitting A. gandensis.
- L21-31. Consider rephrasing: "The genus Acinetobacter (citation) is a member of the family Moraxellaceae, with its type species, Acinetobacter calcoaceticus, which was isolated and described by W. Beijerinck in 1910 under the name Micrococcus calcoaceticus [1]."
Answer: revised as suggested (now L29-31)
- L31-33. Consider rephrasing: "Currently, 80 species with correct, validly published names are assigned to the genus Acinetobacter (https://lpsn.dsmz.de/genus/acinetobacter; date of access: 26 October 2023) [3]." Note that now, there are 81 such species, with the newest A. higginsii (PMID: 37889259) not included in the quoted list yet; but that can be ignored, as this new species is distinct from A. thutiue.
Answer: revised as suggested
- Fig. 1. I can hardly see a reason to include Fig. 1 (16S rRNA) in the main text given its limited value, inferiority to Fig. 2 and partial discordance with Fig. 2. It can be included in the supplementary data file if the authors consider it relevant. Alternatively, the authors can prepare a new tree with all members of the haemolytic clade and discuss differences; but I do not see its usefulness given the well-known unsatisfactory resolution of the 16S rRNA gene below the genus level.
Answer: moved Fig. 1 to the supplementary materials as Fig. S6, as suggested
- L71, 199, 208, Table 1. A. gandensis is neither a member of the haemolytic clade nor a “closely related” species (see Fig. 2) and should be omitted and replaced by missing species of the haemolytic clade (see point 1 above).
Answer: replaced as suggested

This manuscript is a resubmission of an earlier submission. The following is a list of the peer review reports and author responses from that submission.
Round 1
Reviewer 1 Report
Comments and Suggestions for Authors
The paper proposes two novel species names for two soil isolates based on their phenotypic and chemotaxonomic features. However, based on their genomic characteristics, the two isolates represent a single species rather than two. This species is a new member of the hemolytic clade, with A. parvus being its closest relative. The authors' dubitable interpretation would lead to nomenclatural bias and the need for correction. Even though the manuscript has a scientific potential (given the taxonomic uniqueness of the novel strains), it suffers from severe methodological and interpretation drawbacks. The authors are encouraged to consider the following suggestions to improve their information. They are also recommended to read the most recent papers on the Acinetobacter taxonomy, including the recent chapter in the Bergey's Manual of Systematics of Archaea and Bacteria (https://onlinelibrary.wiley.com/doi/10.1002/9781118960608.gbm01203.pub2). No supplementary data were provided and thus could not be reviewed.
Major comments
1. The two novel strains represent a single species. The high ANI (96.05%) and dDDH (65.7%) values strongly indicate the conspecificity of the two strains. In general, ANI values of >95% are considered as indicator of conspecificity, while only ANI values of <93% can reliably indicate of non-conspecfitity. Moreover, the intraspecies ANI and dDDH values of 95-96% and 60-70%, respectively, have been found typical for several Acinetobacter species. Therefore, I disagree with the authors' technocratic application of the ANI and dDDH approximate thresholds (lines 291-295). Further, their conclusion remark (lines 311-319), which explains the rationale for two nomenclatural proposals is scientifically unacceptable, because neither the origin nor a few phenotypic differences (which can reflect intraspecies variations and some of which are highly questionable – see point 5 below) cannot substantiate the taxonomic difference at the species level between genomically conspecific strains.
2. Genus-wide comparative analysis and core-genome phylogenetic reconstruction has to be applied. The authors used 16S rRNA gene- and rpoB-based approaches to identify the phylogenetically closest relatives of their strains, whereas only small subset of strains was analysed using core genome analysis. However, the context-wide phylogenomic analysis of the whole genome sequence of a new bacterium is a crucial component of the current taxonomy, with core-genome analysis being the gold standard and superior to approaches based on single gene markers. In Acinetobacter, for which the genome sequences of all known validly named species are available, genus-wide core-genome comparative analysis is the method of choice for the assessment of the phylogenetic position of novel organisms and for the selection of its phylogenetic neighbors for further taxonomic analyses. Therefore, the authors should provide the results of genus-wide core genome comparative analysis, whereas rpoB/16S rRNA sections can be omitted or moved to the supplementary data section.
3. Single-strain-species-description (SSSD) is discouraged. Description of a general category (species) based on a single individual (strain) is in principle meaningless, providing no reliable information about species-specific or diagnostic traits. That is why the SSSD practice is currently discouraged by taxonomic journals, such as the IJSEM or Syst Appl Microbiol. This drawback is solvable here by the fact that the two isolates are conspecific but differ from each other at the strain level of resolution. Thus, the description of a single novel species can be based on two strains.
4. Comparison of the phenotypes of phylogenetically closest related species. The selection of the strains phylogenetically most related to the new organisms is partially incorrect being apparently based on the results of 16S rRNA gene analysis. Based on the core genome-based phylogeny of the genus, the relevant species are A. parvus and other members of the hemolytic clade, but not A. tandoii, A. indicus or A. bereziniae.
5. Comparative phenotypic analysis of a novel taxon should include mainly properties that are taxonomically relevant (potentially discriminatory at the species level) for the genus. The authors based their phenotypic analysis only on commercial systems API systems, which mostly include tests taxonomically irrelevant for the genus Acinetobacter or on tests useless for non-fermentative bacteria (e.g. Voges-Proskauer, methyl red). Meaningless is testing for anaerobic phototrophy or bacteriochlorophyl (Table 2). The authors may take advantage of the availability of a dataset of the metabolic and physiological features of nearly all known Acinetobacter species based on a standardized, genus-specific phenotypic system based on the original scheme of Bouvet and Grimont (apps.szu.cz/anemec/Phenotype.pdf).
6. Ability to growth anaerobically (line 204) is extremely unlikely, given the strictly aerobic metabolisms of the whole genus. If so, it would be the first known strain of Acinetobacter with such metabolism and the authors should formally amend the description of the genus.
7. Data shown in Table 1 are not believable. E.g.: acetate is assimilated by all known species included in the table; A. parvus does not assimilate lactate; neither mannose nor sorbitol can be used by acinetobacters as carbon source. Data for the last two species are missing.
8. Reviewer's recommendations. The two strains are – based on core genome analysis - phylogenetically closest related to A. parvus, which forms a separate branch within the hemolytic clade with no clearly closer relatives. The two new strains should be compared primarily to A. parvus represented by multiple strains (published data are sufficient) or possibly also to some more members of the hemolytic clade (e.g. metabolically moderately (A. modestus and A. tjernbergiae) using taxonomically relevant tests.
Further comments
9. How “optimal growth” was defined? It is e.g. very unlikely that a single strain of A. parvus grows optimally in a range as wide as 20-35°C (Table 2). Did the authors calculate growth rates?
10. It is not explained how dissimilative denitrification (lines 194 and 354) was tested.
11. The relevance of the antimicrobial susceptibility section is taxonomically questionable. If used, it should be based on standardized methodologies such as CLSI, with the explicit definition of the resistant and susceptible categories. The selection of vancomycin and cycloheximide for Acinetobacter is meaningless, whereas some antimicrobial classes primarily effective against acinetobacters (PMID: 21793988) are missing.
Comments on the Quality of English LanguageThe text contains an unacceptably high number of textual, stylistic and typographic errors.
Reviewer 2 Report
Comments and Suggestions for Authors
"Classical" paper on the description of new bacterial species. Generally well written with few remnants? of previous papers and small methodological discrepancies.
Specific comments
L147 - correct - maxi-mum-parsimony, Mega
L153 - Region 1 spans nucleotide positions 2916–3267 - in which sequence?
L183 - 185 - This section may be divided by subheadings. It should provide a concise and precise description of the experimental results, their interpretation, as well as the experimental conclusions that can be drawn????
L201 - VNH17T was growth
L204 - dtto
Table 1 - the data shown in table for VNH17 and VNK23T isolates are different from those in text - source of isolation, cell size colony color????
Most of strain you used for comparison were isolated from acidic environments. Could you provide the data on pH of oil-contaminated soils used for the isolation of your strains?
L261 - we further genome features analysis??
Figure1, Figure2 - why different sets of sequences were used for the tree construction? Why different types of tree are shown? rooted versus unrooted? Please add the Genbank accession numbers of VNH17 and VNK23T sequences
Comments on the Quality of English Language
Generally well written manuscript with few typing errors methodological discrepancies.
Reviewer 3 Report
Comments and Suggestions for Authors
In the presented manuscript, Nhan Le Thi Tuyet and Jaisoo Kim, presented “Acinetobacter thutiue sp. nov. and Acinetobacter kontum sp. 2
nov., isolated from motorbike repairing workshop soil ”.
The manuscript is suitable for publication in diversity, the language should be revised and additional analyses should be performed to present a comprehensive analysis as proposed by the authors.
I provide some major and minor comments below.
Major;
1. Authors should perform a MLSA with at least 5 housekeeping genes. In order to obtain a more robust phylogeny. Compare the topology of the tree with UBCG (concatenated alignment of 92 core genes).
2. All phylogenetic trees must include the proposed name, collection and accession number.
3. Perform a heatmap with the results of ANI and isDDH, the graph is more suitable to understand the evolutionary differences of the species of the genus.
4. Perform in silico phenotyping analysis.
Minor,
The writing and grammar of the article must be revised.
1. Send 16S phylogenetic tree to supplementary material.
2. Line 284. The Rast. In uppercase.
Comments on the Quality of English LanguageModerate editing of English language required